# Anti-Dengue Activity of Lipophilic Fraction of *Ocimum basilicum* L. Stem

**DOI:** 10.3390/molecules28031446

**Published:** 2023-02-02

**Authors:** Rajesh Kumar Joshi, Shivankar Agarwal, Poonam Patil, Kalichamy Alagarasu, Kingshuk Panda, Sarah Cherian, Deepti Parashar, Subarna Roy

**Affiliations:** 1ICMR-National Institute of Traditional Medicine, Belagavi 590010, Karnataka, India; 2ICMR-National Institute of Virology, 20-A, Dr. Ambedkar Road, Pune 411001, Maharashtra, India

**Keywords:** dengue, chikungunya, antiviral, *Ocimum basilicum* L. Merr., phytosterols

## Abstract

*Ocimum basilicum* L. is used to cure many types of fever in traditional medicine. This study aims to explore the antiviral activity of the lipophilic fraction of the stem of *O. basilicum* (LFOB) against dengue virus (DENV) and chikungunya virus (CHIKV). The LFOB was analyzed using GC-FID and GC-MS. The antiviral activity of LFOB was studied using the Vero CCL-81 cell line. The cytotoxicity assay was performed using 3-(4,5-dimethythiazol-2-yl)-2,5-diphenyl tetrazolium bromide (MTT). In vitro antiviral activity and FFU assay were used to determine and confirm antiviral activity against DENV and CHIKV. Twenty-six compounds were identified in LFOB using GC/MS. The most abundant compounds were *β*-sitosterol (22.9%), stigmasterol (18.7%), and campesterol (12.9%). Significant reduction in DENV titre was observed under pre- and post-infection treatment conditions at a concentration of 3.125 µg/mL, but no anti-CHIKV activity was observed. Our earlier and the present AutoDock-Vina-based in silico docking study revealed that *β*-sitosterol and stigmasterol could form strong interactions with the DENV E glycoprotein and DENV RdRp domain, respectively. Our findings suggest that LFOB can inhibit DENV infection and might act as a potent prophylactic/therapeutic agent against DENV-2. In silico results suggested that *β*-sitosterol and stigmasterol may block the viral entry by inhibiting the fusion process and viral replication respectively.

## 1. Introduction

In the current era of drug confrontation and adverse outcomes of synthetic drugs, widespread attention on herbal remedies and metabolites of plant extracts has increased among investigators and the general population globally. *Ocimum basilicum* L. (basil) (Lamiaceae) has traditionally been considered as an important herb. More than 150 varieties of this genus have been identified, and basil is considered as an essential commercial product worldwide [1]. In traditional African medicine, *O. basilicum* is used to cure many types of fever and whooping cough [2]. Basil has been used in traditional medicine to treat anxiety, diabetes, heart disease, headache, neurological pain, and neurodegenerative disorders and a an anti-convulsant [3]. Additionally, seeds have been used to treat inflammatory bowel disease [4]. In Unani medicine, the seed part is considered de obstruent, flatus-relieving, and restorative. Due to its mucilaginous qualities, it is highly regarded when consumed whole; but when crushed, it is believed to be astringent [5]. Seeds are mucilaginous and helpful for internal piles, gonorrhoea, oliguria, chronic diarrhoea, epilepsy nephritis, and dysentery [6,7]. In order to lower the risk of disease, it is one of the most often utilised medicinal plants in Morocco which is associated with plasma cholesterol and atherosclerosis treatment [8] and preclusion of heart disease [9]. Traditionally, basil is used as a mosquito and housefly repellent in part of western Spain [10] and Cameroon [11].

Basil has excellent nutritional value and phytochemical content. It consists of vitamin B complex (B1, B2, B3, B5, B6, B9) and choline. Minerals are present in high quantities along with numerous classes of secondary metabolites which broaden its range of effects [12]. Basil has been shown to have antibacterial, antioxidant, anti-inflammatory, anti-cancer, mosquito larvacidal, anti-parasitic, lipidemic, cytoprotective, immunomodulatory, and anti-convulsant qualities [13,14]. Extensive research has been carried out on linalool, the principal constituent of essential oil of *O. basilicum*, followed by methyl eugenol, methyl chavicol [15,16,17,18]. However, researchers have observed diverse variations in oil constituents of *O. basilicum* due to geographical area [16]; hence, it is considered to be polymorphic [19]. Abundant literature is available on both extracts and purified compounds of basil against various viruses. Research on crude and ethanol extract of *O. basilicum* showed a vast range of antiviral activity against adenoviruses, hepatitis B, enterovirus 71 coxsackievirus B1, and herpes viruses. Essential oil obtained through hydro-distillation showed significant in vitro antiviral activity against herpes simplex virus 1 (HSV). Further, it has been reported that the organic solvent extracts of *O. basilicum* showed inhibition against diverse phases of the HSV viral growth cycle [20]. The components from *O. basilicum*, viz., ursolic acid, showed antiviral activity against human immunodeficiency virus 1 (HIV-1) [21,22]. The linalool and apigenin from *O. basilicum* displayed vigorous antiviral activity in highly diluted ethanolic basil leaf extract, efficiently inhibiting the Zika virus and preventing its entrance into the host cell without affecting the cells. Phytoconstituents of *O. basilicum* are responsible for the effective antiviral activity and are appropriate drug candidates for treating or managing viral infection [22,23]. This communication presents the antiviral activity of the lipophilic fraction of the stem of *O. basilicum* (LFOB) against dengue (DENV) and chikungunya (CHIKV).

## 2. Results

### 2.1. Chemical Constituents of LFOB

The compounds of the LFOB were identified according to their retention indices and mass spectra, containing 95.4% of the overall oil ingredients. The identified components are listed in Table 1, along with the quantity (%) based on the GC-FID peak areas of each component and its retention index. Twenty-six compounds were identified in the LFOB, from which the most abundant compound was *β*-sitosterol (22.9%), followed by stigmasterol (18.7%) and campesterol (12.9%). The other identified compounds (<10%) were hexadecanoic acid (7.5%) and linolenic acid (6.8%) (Appendix A). LFOB was found to be rich in triterpenoids and sterols (56.0%), followed by long-chain hydrocarbons (18.2%), long-chain oxygenated hydrocarbons (14.8%), chromane terpenoid (3.9%), phenylpropanoids (2.3%), oxygenated mono- and sesquiterpene (0.1%) and sesquiterpene hydrocarbon (trace) type compounds.

### 2.2. Effect of LFOB Treatment on Proliferation of Vero Cells (MTT Assay)

The effect of the LFOB on Vero cell viability was investigated using the MTT assay. The cell viability was greater than 90% at concentrations ≤3.125 µg/mL. Above 3.125 µg/mL concentration, dose-dependent toxicity was observed (Figure 1). The CC50 value was 10.67 µg/mL (Figure 1). Concentrations of 3.125 μg/mL were used in further experiments since it was not toxic to the cells.

### 2.3. Primary Screening of LFOB against DENV and CHIKV Replication

The LFOB was examined under pre- (for prophylactic effect), co- (for virucidal effect), and post-treatment (for therapeutic effect) conditions to determine if it exhibits antiviral effects towards DENV and CHIKV at a non-toxic dose (3.125 μg/mL). Focus forming unit (FFU) assay was used to determine the effect of the LFOB on the virus. At 3.125 μg/mL concentrations, the LFOB significantly reduced DENV titres under pre-treatment and post-treatment conditions (Figure 2a). The LFOB did not affect DENV titres under co-treatment conditions (Figure 2b). The LFOB had no anti-chikungunya activity, as assessed via FFU assay under pre-, co-, and post-treatment conditions (Figure 2b).

### 2.4. Antiviral Activity of LFOB against DENV

The anti-dengue effect of the LFOB was assessed at different concentrations under pre-treatment and post-treatment conditions since it showed anti-DENV activity at the highest non-toxic concentrations. The post-treatment of cells with 3.125 µg/mL of the LFOB showed a significant reduction in virus foci (mean virus log_10_ titre 4.141 FFU/mL) compared to virus control (5.354 mean virus log_10_ FFU/mL value) (*p* < 0.0001) (Figure 3). A non-significant reduction in virus titre was observed at a concentration of 1.56 µg/mL.

### 2.5. In-Silico Interaction Studies of Compounds with DENV Protein Targets

A computational docking study in AutoDock Vina was performed to explore the potential mechanisms of action of constituents of the LFOB except *β*-sitosterol on DENV proteins. The AutoDock Vina algorithm was used to compute the binding energy. *β*-sitosterol was excluded from docking analysis since our earlier in-silico study has shown that *β*-sitosterol binds to DENV E glycoprotein with high affinity [24]. After initial screening, all the identified compounds in the LFOB were ranked based on their binding energy. Two compounds, stigmasterol and campesterol, showed the highest affinities for DENV targets. A further study was conducted to determine the best position for these compounds to bind to the target protein active site.

The compound stigmasterol showed the highest binding affinity with DENV NS1 antigen, and interaction studies demonstrated that the compound has a binding affinity of −8.3 kcal/mol. Analysis of the stigmasterol docked complex showed an unfavorable donor bond with LYS 227. Van der Waals interaction was observed with GLU 154, GLU 173, ASP 180, SER 181, ASP 176, SER 228, and ASN 234 in the stigmasterol-NS1 docked complex. Few alkyl and Pi-alkyl interactions were observed with LYS 172, PHE 178, PRO 226, TRP 232, LEU 206, TRP 210, and LYS 182. No hydrogen bond formation was observed in the docking analysis, but the compound was bound to a functional residue.

The compound campesterol showed the highest binding affinity against E glycoprotein. The binding affinity of campesterol with DENV E glycoprotein was observed as −8.2 kcal/mol. The docking interface of campesterol with DENV E glycoprotein showed a single conventional hydrogen bond with ARG 350. Though the binding affinity score was high upon interaction with campesterol and DENV E glycoprotein, the interacting residues were not in the functional sites in the DENV E glycoprotein.

The compound stigmasterol showed the highest binding affinity with DENV NS3 helicase, and interaction studies revealed that the compound has a binding affinity of −8.9 kcal/mol. Several Van der Waals interactions were formed with SER 364, MET 537, and VAL 544. Few alkyl interactions were observed with LEU 443, ARG 387, ARG 599, PRO 291, and PRO 543. None of the interactions were observed in any functional residues of the NS3 helicase domain.

The compound stigmasterol showed the highest binding affinity with DENV NS2B-NS3 protein, and interaction studies revealed that the compound has a binding affinity of −8.1 kcal/mol. A Van der Waals interaction with PHE 1130 was observed in the docked complex of stigmasterol and NS2B-NS3 protein. Additionally, a few Pi-alkyl bonds were formed with PRO 1132, LEU 1128, TRP 1050, TYR 1161, and HIS 1051 in the docked complex with stigmasterol. The compound thus interacted with residues of both catalytic and allosteric site at their best docked position and hence may not form a stable complex.

Both compounds showed a high binding affinity with DENV-RdRP. Both the compounds stigmasterol and campesterol showed a binding affinity of −8.4 kcal/mol. Interestingly, stigmasterol is the only compound that binds to a potential binding site near the catalytic site, interacting with the residue of all three conserved motifs (Q598-N614, G662-D664, and C709-R729) as well as residues of the priming loop (H786-M809). The interaction shows a single hydrogen bonding with Arg 737 and the hydroxyl group of stigmasterol. An unfavorable donor–donor interaction also formed with Lys 460. There were multiple hydrophobic interactions, including an alkyl and Pi-alkyl interaction, also noted with Tyr 607 (motif I), Cys 709 (motif II), Trp 795Ile, 797 (priming loop). Except this, few Van-der Waals interactions were observed with His 798, Ser 796, Thr 794, Ser 710, Arg 729, Glu 459, Asn 610, Gly 662, Asp 663, Lys 457, Gln 603, Thr 606, and Ser 661 (Figure 4).

The compounds stigmasterol and campesterol showed the highest binding affinity with DENV NS5 methyltransferase protein. Few Van der Waals and hydrophobic interactions were observed in the docked complex. No hydrogen interactions were observed upon interaction. Both compounds showed a binding affinity of −7.4 kcal/mol. However, the interacting residues were not in the functional sites in the NS5 methyl transferase domain.

## 3. Discussion

Even with recent advancements in medical technology, dengue therapeutics face a significant challenge. The race to get effective and safe dengue therapeutics continues to be an imperative necessity. Ethnopharmacology has substantially contributed to developing new therapeutics [25]. Recently, medicinal plants and their bioactive metabolites have become a significant focus of interest for developing effective and affordable drugs against various viral infections. *O. basilicum* is known for various medicinal uses. This study evaluated the antiviral activity of the LFOB against DENV and CHIKV. Cytotoxicity assay revealed that concentrations >3.125 μg/mL were toxic to the cells, suggesting the presence of toxicity-inducing chemicals in the LFOB. The toxicity was higher compared to similar extracts from other plants such as *Sauropus androgynus* [24]. Lipophilic fraction of *S androgynus* leaves prepared using hexane is known to contain *β*-sitosterol and eleven other compounds [25], while the LFOB from *O. basilicum* contains stigmasterol and campesterol in addition to *β*-sitosterol with 23 other compounds which might have contributed to cytotoxicity even at lower concentrations. Several lipophilic compounds also showed antiviral activity against DENV [26,27,28,29,30].

The in vitro antiviral assays of the extract revealed that at a concentration of 3.125 μg/mL, the extract significantly reduced DENV titres under pre-treatment and post-treatment conditions. The antiviral activity before and after infection suggests the extract might affect the host cell receptor and might also act on viral proteins. To find out the probable compound in the extract that interacts with viral proteins, docking analysis was performed for all the 26 compounds present in the extract with various DENV non-structural and structural proteins. The docking study for the initial screening that was undertaken using 25 compounds in the LFOB (as indicated in Table 1) against DENV proteins revealed that stigmasterol and campesterol showed maximum binding scores against different DENV target proteins. Our earlier study showed that *β*-sitosterol interacted strongly with DENV E glycoprotein [24]. Notably, these three compounds (stigmasterol, campesterol, and *β*-sitosterol) were the most abundant in the leaf extract, totally amounting to 54.5%.

The in vitro finding shows that the LFOB can inhibit the DENV under pre-treatment conditions. One major compound, *β*-sitosterol, present in the LFOB, could block the immune responses mediated by RIG-I signaling and deleterious IFN production that could help to reduce the viral load [31]. Furthermore, it has also been reported that it interferes with viral gene expression and inhibits viral replication by interfering with viral immune signaling, hijacking pathways [32]. As of now, there are no reports available on *β*-sitosterol, stigmasterol and campesterol targeting DENV proteins. Hence, in the present study, we used the blind docking protocol to understand the likely binding sites. DENV inhibition on pre-treatment raises the possibility of interacting with a cellular receptor responsible for DENV binding in Vero cells.

The in vitro post-infection inhibition proposes that the compounds present in the LFOB could interfere with structural/non-structural proteins in the DENV genome. Based on the docking results, we made two hypotheses for the antiviral effect in post-infection. This inhibition under post-treatment conditions could be due to the binding of the *β*-sitosterol compound in the LFOB to the virus surface E glycoprotein [25]. Based on the previous report, amino acids THR-48, ALA-50, GLN-200, ILEU-270, and THR-280 are critical for membrane fusion [33]. Our studies also revealed that *β*-sitosterol binds to the amino acids, which are critical for membrane fusion [24]. The suppression of the fusion process may therefore cause a decrease in viral foci after post-treatment. Viral RdRP is a major target for developing any antiviral drug. There are three subdomains within the RdRp domain: thumb, fingers, and palm. A conformational change is hypothesized to occur during the de novo initiation of RNA synthesis in the thumb subdomain, which contains the priming loop [34]. The docking study of stigmasterol shows that it interacted at the RNA binding site, specifically at the three conserved motifs, motif B (Q598-N614), motif C (G662-D664), and motif E (C709-R729), and the residues of the priming loop (H786-M809). The compound formed a single hydrogen bond with the amino acid Arg-737, a strictly conserved residue known to initiate replication using de-novo synthesis [35]. These interactions suggest that the compound stigmasterol could inhibit the de-novo RNA synthesis and elongation of RNA.

In vitro and in vivo effects of *β*-sitosterol, stigmasterol, and campesterol on DENV infection and replication needs evaluation and might help in identifying the compound responsible for the anti-DENV activity. Apart from these three compounds, hexadecanoic acid, linolenic acid and *n*-untriacontane were reported at higher concentrations (~5% each) in the LFOB, and the antiviral activities of these compound also need evaluation and might explain the antiviral activity during pre-treatment. Our recent study indicated tht the lipophilic fraction of *S. androgynus* leaves exerted anti-DENV activity similar to the LFOB; however, both fractions differ with regard to the chemical composition. The lipophilic fraction of *S. androgynus* leaves had high concentrations of squalene, vitamin E, *δ*-tocopherol, *γ*-tocopherol, *β*-sitosterol, and hexadecanoic acid [25]. The common compounds between the two extracts in higher concentration are *β*-sitosterol and hexadecanoic acid, suggesting that other compounds in the LFOB might have contributed to higher toxicity as well as antiviral activity. Identying the compounds specific to LFOB with low toxicity but effective antiviral activity might help in formulating phyto pharmaceutical drugs against dengue which can be further evaluated in in vivo studies.

The LFOB did not exert substantial antiviral activity against CHIKV suggesting that chemicals in the extract may not be in sufficient concentration to exert activity or do not posses anti-CHIKV activity. The same has been observed for the lipophilic fraction of *S. androgynus* leaves, suggesting that lipophilic extracts are not potential candidates for antiviral screening against CHIKV.

## 4. Materials and Methods

### 4.1. Collection and Identification of Plant Material

The fresh stem of *O. basilicum* was collected in October 2021 from Belagavi, Karnataka, India. Identification of plant was done by Dr. H. V. Hegde, Taxonomist at ICMR-National Institute of Traditional Medicine (NITM), Belagavi (Specimen no. RMRC-532).

### 4.2. Preparation of Extract

The stems of *O. basilicum* were cut into small pieces (approximately 01 to 1.5 cm long), shade dried, grounded to powder, and sieved with 50 mesh size. The 20 g of powder was extracted from Soxhlet using hexane for 6 h. The hexane-soluble part was evaporated in a rotary evaporator at 25 °C. The yield of lipophilic fraction of stems of *O. basilicum* (LFOB) was 1.7% (*w*/*w*), and it was kept in a sealed amber color vial at −4 °C until analysis.

### 4.3. Analysis of the Extract

The quantitative and qualitative analysis and identification of the constituents of LFOB (1% solution dissolved in *n*-hexane) was carried out with a repoon Varian 450 gas chromatograph (GC) used with Zebron ZB-5-Phenomenex fused silica capillary columns (30 m × 0.25 mm diameter, 0.25 mm film thickness) fitted with a fused silica capillary column. The oven temperature programmed with linear mode went from 60 °C to 280 °C at a 3 °C/min ramp; using nitrogen (1.0 mL/min flow rate), for the injector and detector, temperatures were fixed at 290 °C and 300 °C, respectively. The injection volume was 1.0 μL of 1% solution, and the split ratio was set at 1:50 [30,31,32]. The ITQ 1100 interfaced Trace Ultra Gas Chromatography-Mass Spectroscopy (GC-MS) (Thermo Scientific, Waltham, MA, USA) was employed for qualitative analysis with the column mentioned above and GC oven temperature program using helium as a carrier gas at 1.0 mL/min. The injector temperature and injection volume were 290 °C and 1.0 μL, respectively; split ratio 1:50. The mass scan range for the MS spectrum was 50–650 amu at 70 eV [36,37,38,39,40]. Identification of constituents was made using the supported retention index (RI) regarding the homologous series of *n*-alkanes C_8_-C_25_ including identical experimental conditions on the ZB-5 column, a NIST and WILEY MS library search, and via examination with the MS literature information [41] and co-injection of economic samples with ≥98% purity. Without using a correction factor, the relative concentrations of each molecule were determined based on the GC’s peak region of FID response.

### 4.4. In Vitro Antiviral Activity

#### 4.4.1. Cell Culture and Virus Stock

The Vero CCL-81 cell line was maintained in minimal essential media (MEM) (Himedia, Kennett Square, PA, USA), supplemented with 10% FBS (Gibco, US Origin) and Antibiotic-Antimycotic (Sigma-Aldrich, St. Louis, MO, USA) at 37 °C and 5% CO_2_. Dengue (DENV) Serotype-2 (Strain no. 803347) and chikungunya (CHIKV, Strain no. 061573, P-2, African genotype) were used for this study. DENV-2 stock was prepared in C636 (mosquito cell line), and CHIKV stock was propagated in Vero cells and stored at −80 °C.

#### 4.4.2. Cytotoxicity and Cytopathic Effect Inhibition Assay

The cytotoxicity assay was performed using 3-(4,5-dimethythiazol-2-yl)-2,5-diphenyl tetrazolium bromide (MTT) to determine the effect of LFOB on Vero CCL-81 cells [42]. Briefly, a series of Vero cell monolayers were incubated with different concentrations of the LFOB (0 to 200 g/mL) for five days at 37 °C, followed by 3 h of incubation with MTT (5.0 mg/mL). The solubilized formazan crystals were measured using a microplate reader (BioTek Synergy, Agilent, CA, USA) at 570 nm. The concentration at which the LFOB showed 50% toxicity (CC_50_) values was calculated via nonlinear regression analysis using GraphPad Prism software version 7.0.

#### 4.4.3. Antiviral Activity of the LFOB in the In Vitro System

The LFOB was studied for its antiviral activity under pre-treatment, co-treatment, and post-treatment conditions. During pre-treatment, different concentrations of LFOB were applied to the cells for 24 h at 37 °C, and the culture supernatant was removed. The cells were infected with 0.1 multiplicity of infection (MOI) of DENV-2 or 0.01 MOI for CHIKV and incubated at 37 °C for 1 h. The unbound virus particles were removed by phosphate-buffered saline (PBS), washed twice, and incubated with maintenance media. During co-treatment, the virus with different concentrations of the LFOB was incubated for one hour and was used to infect cells for one hour.

For post-treatment, the cells were infected with 0.1 MOI DENV-2 or 0.01 MOI for CHIKV for 1 h and treated with the LFOB immediately after infection. For all treatments, the plates were incubated for five days in the case of DENV-2 and two days for CHIKV after infection. After incubation, the plates were freeze-thawed and centrifuged to collect culture supernatant for estimation of virus titre via FFU assay. All the experiments were performed in triplicates. For significant results, the experiments were repeated in triplicates.

#### 4.4.4. FFU Assay

To determine the titres of DENV-2 and CHIKV, FFU assays were performed as described earlier [34,43].

### 4.5. Docking Studies Using Viral Proteins and Molecular Modeling of Chemical Compositions

The GC-MS analysis confirmed the presence of twenty-six different compounds in the LFOB. The two-dimensional structure of all the compounds was retrieved from the PubChem database. We retrieved three-dimensional (3D) structures from the Protein Data Bank (PDB) of every available DENV-2 target protein, including the DENV-2 NS1 crystal structure (4O6B), DENV-2 envelope glycoprotein (1OKE), DENV-2 NS2B-NS3 protease (4M9K), NS3 helicase domain (2BHR), NS5 methyltransferase domain (1R6A), and NS5 RdRP domain (5ZQK) (https://www.rcsb.org/ (accessed on 1 November 2022)). Using AutoDock Vina, all DENV-2 target crystal structures were docked with their ligands. In the initial processing of the proteins, polar hydrogen and gasteiger charges were added using AutoDockTools (ADT). The docking study was conducted with default parameters and values. In our first virtual screening, we selected all chemical compounds as ligands and independently calculated their binding affinity against the target protein. The grid box is set to search all over the protein for binding sites. Following the initial docking evaluation, compounds with maximum scores were selected for further docking evaluation, such as binding site prediction and interaction analysis. BIOVIA Discovery Studio 2020 was used to analyze all docked complexes.

### 4.6. Statistical Analysis

The viral RNA/FFU titre or percent infected cells were compared between the different groups using one ANOVA. A *p* value of less than 0.05 was considered as significant. The statistical analysis was done using GraphPad prism software version 7.0.

## 5. Conclusions

Iro antiviral activity under pre- and post-treatment circumstances showed that LFOB can inhibit DENV infection and may thus be an effective preventative or therapeutic drug for DENV-2. The unfavorable results against CHIKV suggest that lipophilic chemicals are less efficient at low concentrations and provide evidence that the LFOB may not be useful against specific viruses. The LFOB components could interact with the DENV E protein by obstructing the fusion process, potentially preventing viral entry or release and also with the DENV RdRp domain, inhibiting the viral replication. Thus, the use of *O. basilicum* in traditional medicine as attributed to alleviating fever conditions is also understood to target DENV proteins as observed in the present study and needs further in vivo validations.

## Figures and Tables

**Figure 1 molecules-28-01446-f001:**
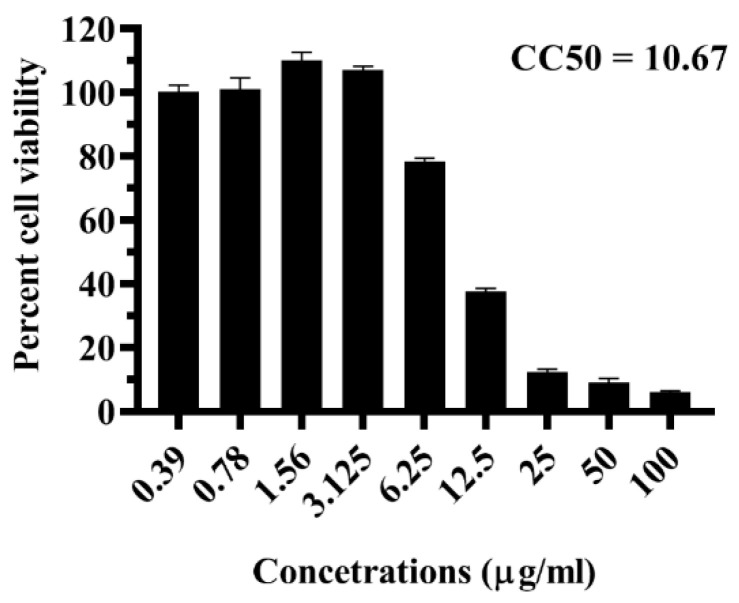
Effect of LFOB on cell viability as measured via MTT assay at two independent time points, the experiments were conducted in triplicate.

**Figure 2 molecules-28-01446-f002:**
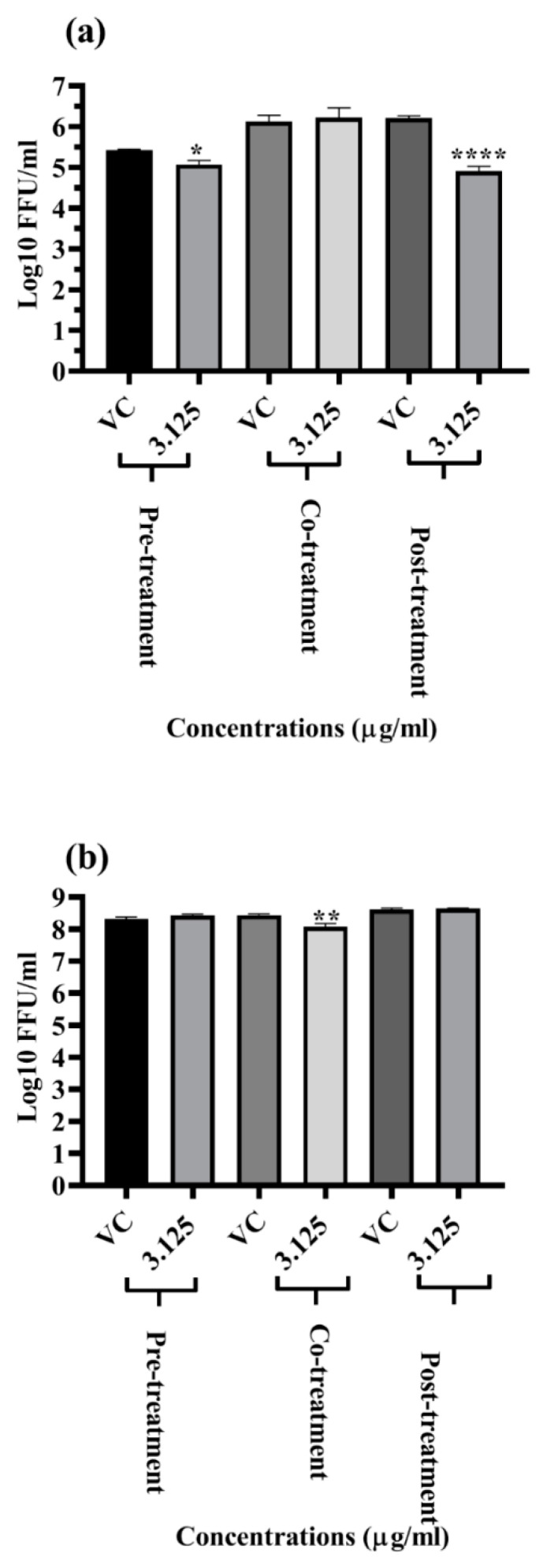
Antiviral screening of LFOB at maximum non-toxic concentration against (**a**) DENV and (**b**) CHIKV under pre-, co-, and post-treatment conditions. All the treatment conditions were compared with the virus control. **** *p* value < 0.0001, ** *p* < 0.01, * *p* < 0.05 vs. control.

**Figure 3 molecules-28-01446-f003:**
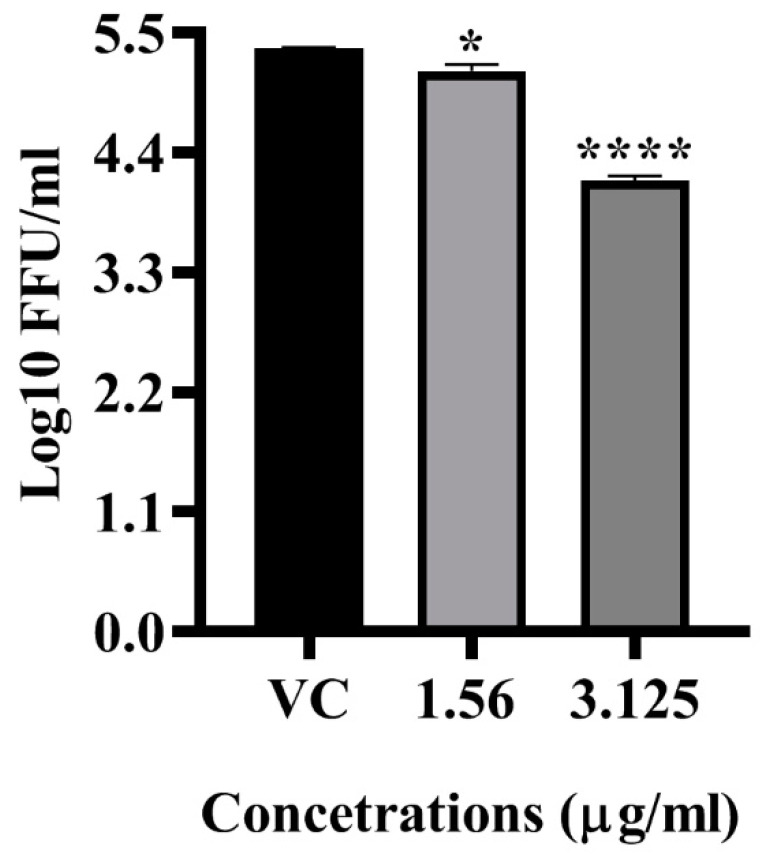
Dose dependent effect of LFOB on DENV-2 as assessed via focus forming unit assay. All the values are expressed as mean ± SD of three experiments. **** *p* < 0.0001; * *p* < 0.05 vs. control.

**Figure 4 molecules-28-01446-f004:**
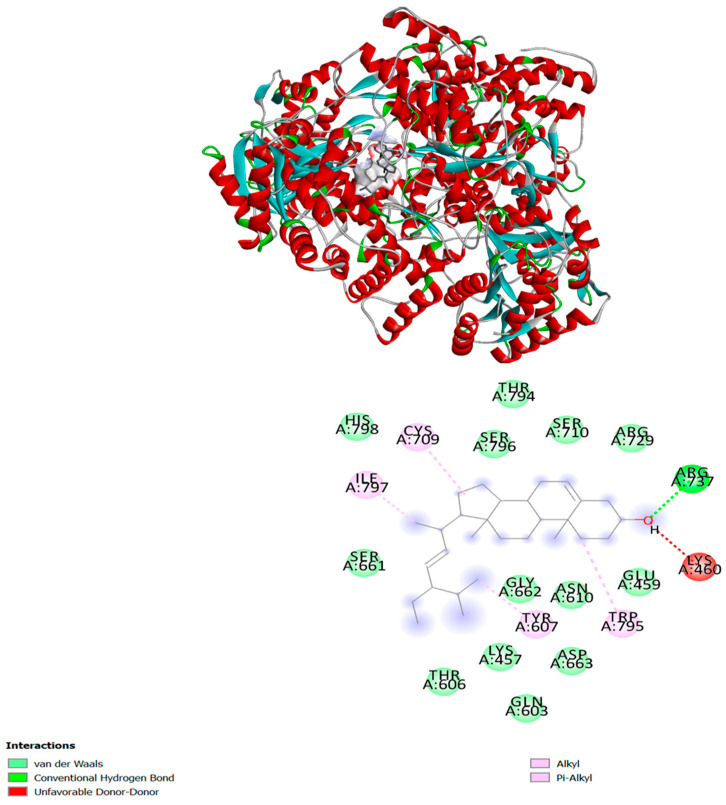
Molecular interaction of stigmasterol with DENV RdRP domain. The different types of interaction are indicated by different colors. The interactions are visualized and analyzed via BIOVIA Discovery Studio 2020.

**Table 1 molecules-28-01446-t001:** Chemical composition of *Ocimum basilicum* stem extract.

Compound	RI	% Content	Identification
Terpinene-4-ol	1173	t	RI, MS, CI
*α*-Terpineol	1191	0.1	RI, MS, CI
Methyl chavicol	1198	0.5	RI, MS, CI
Chavicol	1262	0.2	RI, MS
(*E*)-Anethole	1288	t	RI, MS, CI
Eugenol	1361	0.1	RI, MS, CI
*β*-Caryophyllene	1421	t	RI, MS, CI
*n*-Hexadecane	1600	0.2	RI, MS, CI
*epi*-*α*-Bisabolol	1686	0.1	RI, MS
Benzyl benzoate	1766	1.5	RI, MS
Hexadecanoic acid	1976	7.5	RI, MS, CI
Methyl linoleate	2114	0.5	RI, MS
Linolenic acid	2154	6.8	RI, MS
*n*-Tricosane	2300	0.5	RI, MS, CI
*n*-Pentacosane	2500	0.7	RI, MS, CI
*n*-Heptacosane	2700	1.5	RI, MS, CI
Squalene	2831	1.5	RI, MS
*n*-Nonacosane	2900	1.9	RI, MS, CI
*n*-Triacontane	3000	1.3	RI, MS, CI
*n*-Untriacontane	3100	5.7	RI, MS
Vitamin E	3131	3.9	RI, MS, CI
Dotriacontane	3200	1.9	RI, MS
Campesterol	3219	12.9	RI, MS, CI
Stigmasterol	3256	18.7	RI, MS, CI
*n*-Tritriacontane	3300	4.5	RI, MS
*β*-Sitosterol	3319	22.9	RI, MS, CI
Oxygenated monoterpenes		0.1	
Sesquiterpene hydrocarbon		t	
Oxygenated sesquiterpene		0.1	
Phenylpropanoids		2.3	
Long chain hydrocarbons		18.2	
Long chain oxygenated hydrocarbons		14.8	
Triterpenoids		56.0	
Chromane terpenoid		3.9	
Total identified		95.4	

RI = retention index relative to C_8_-C_25_ *n*-alkanes on ZB-5 column, MS = NIST and Wiley library and the literature, CI = Co-injection of commercial samples, *t* = trace (<0.1%).

## Data Availability

Data are contained within the article.

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
