# Peer review of "Anti-Dengue Activity of Lipophilic Fraction of Ocimum basilicum L. Stem"

_molecules, 2023, doi:10.3390/molecules28031446_

Round 1
Reviewer 1 Report
The manuscript “Anti-dengue activity of phytosterol-rich extract of Ocimum basilicum L.” [molecules-2106367-peer-review-v1] written by Rajesh Kumar Joshi, Shivankar Agarwal, Poonam Patil, Kalichamy Alagarasu, Kingshuk Panda, Mahadeo Kakade, D. Kusuma Sai, Sarah Cherian, Deepti Parashar and Subarna Roy describes an isolation of non-polar (lipophilic) natural products from Ocimum basilicum L. components of this extract are identified by GC-MS and GC-FID. The antiviral activity of the entire extract against dengue and chikungunya are determined and the cytotoxic effect is tested. Furthermore, in silico binding studies of most abundant engrediants β-sitosterol, stigmasterol and campesterol with DENV E glycoprotein and DENV RdRp domain.
All experiments and calculations are performed with relative modern and quite common state of the art methods. The overall work seems relative well planned and performed. However, some of the results and logical connections in the design of experiments can only be understood by the reviewer from the context. Thus there are a few comments about the interpretation of the data that should be dispelled by the authors (see below).
The results gained possess some importance in furthering our knowledge of (lipophilic) natural products (not only) from Ocimum basilicum L. and their antiviral potential, in particular against dengue and chikungunya. However, there are some weaknesses in the presentation and the manuscript hence should be revised (see comments).
The manuscript is of interest in the fields of Pharamcautical Chemistry, Natural Product Chemistry, Phytochemistry as well as to some extent in medicinal Chemistry. However, the reviewer has some comments that should be considered by the authors before the manuscript can be accepted for publication in "Molecules".
Scientific Comments:
a.) The authors chose the three most common compounds of isolation for the in silico studies. This selection should be justified more intensively, also against the background of why the antiviral activity could emanate from these compounds in particular.
b.) The selection of the studied binding sites should be better explained. The possible influences of inhibition/binding to (other) allosteric sites should be discussed in more detail.
c.) The results (spectra) of the GC and MS measurements could be presented in a supplementary material and thus secure the identification.
Comments on the presentation and description
d.) The results of the cytotoxic measurements can be included somewhat more intensively in the results and in the discussion.
e) The further investigations are clearly and also clearly presented in the understanding of the reviewer. However, the "red thread" is somehow missing in the presentation. The authors are therefore encouraged to connect the individual steps more pointedly in the description. Thus, they allow the readers a somewhat better understanding of their train of thought when conducting the study.
Minor comments:
f.) Line 24: Ocimum basilicum in italics
g.) Please use capital “L” for liter throughout the manuscript.
h.) Line 157: “stigmasterol” without capital “S”
j.) Line 168: Two fullstops?
j.) Line 239: “~01” or “~01.”?
k.) Line 270: Please use “°” for the Temperature und not superscript “0”.
l.) Line 275 and also in some other lines: Please use “h” for hours. (Not “hours” or “hrs”.)
Author Response
Response to reviewer’s comments
We appreciate the reviewer’s encouraging, critical, and constructive comments on this manuscript. The comments have been extensive and helpful in improving the manuscript. We strongly believe that the comments and suggestions have increased the scientific value of the revised manuscript by many folds. We have taken them fully into account in the revision. We are submitting the revised manuscript with the suggestion incorporated in the manuscript. The manuscript has been revised per the reviewer’s comments and suggestions. The changes have been made in the manuscript with a different color. Our responses to all the comments are as follows:
Comments and Suggestions for Authors
Reviewer #1: Reviewer's Comments
The manuscript “Anti-dengue activity of phytosterol-rich extract of Ocimum basilicum L.” [molecules-2106367-peer-review-v1] written by Rajesh Kumar Joshi, Shivankar Agarwal, Poonam Patil, Kalichamy Alagarasu, Kingshuk Panda, Mahadeo Kakade, D. Kusuma Sai, Sarah Cherian, Deepti Parashar and Subarna Roy describes an isolation of non-polar (lipophilic) natural products from Ocimum basilicum L. components of this extract are identified by GC-MS and GC-FID. The antiviral activity of the entire extract against dengue and chikungunya are determined and the cytotoxic effect is tested. Furthermore, in silico binding studies of most abundant engrediants β-sitosterol, stigmasterol and campesterol with DENV E glycoprotein and DENV RdRp domain.
All experiments and calculations are performed with relative modern and quite common state of the art methods. The overall work seems relative well planned and performed. However, some of the results and logical connections in the design of experiments can only be understood by the reviewer from the context. Thus, there are a few comments about the interpretation of the data that should be dispelled by the authors (see below).
The results gained possess some importance in furthering our knowledge of (lipophilic) natural products (not only) from Ocimum basilicum L. and their antiviral potential, in particular against dengue and chikungunya. However, there are some weaknesses in the presentation and the manuscript hence should be revised (see comments).
The manuscript is of interest in the fields of Pharamcautical Chemistry, Natural Product Chemistry, Phytochemistry as well as to some extent in medicinal Chemistry. However, the reviewer has some comments that should be considered by the authors before the manuscript can be accepted for publication in "Molecules".
Scientific Comments:
- a) The authors chose the three most common compounds of isolation for the in-silico studies. This selection should be justified more intensively, also against the background of why the antiviral activity could emanate from these compounds in particular.
Response: Thank you for the query. We would like to clarify that as mentioned on page 5 (Lines 118-125), the initial in silico screening was undertaken using AutoDock Vina for all the 26 identified compounds (Table 1). Notably, the three compounds [β-sitosterol (22.9%), stigmasterol (18.7%), and campesterol (12.9%)] which were the most abundant in the leaf extract, showed highest affinities for DENV targets. Thereafter these 3 compounds were considered for in-depth docking studies to determine the best position for these compounds to bind to the target protein active site.
We have now included additional statements in the revised manuscript (discussion section) to justify and make this more clear for the reader.
- b) The selection of the studied binding sites should be better explained. The possible influences of inhibition/binding to (other) allosteric sites should be discussed in more detail.
Response: As of now, there was no report available on β –sitosterol, stigmasterol and campesterol targeting DENV proteins. Hence, in the present study, we used the blind docking protocol to understand the likely binding sites. A statement to this effect has now been included in the revised manuscript, in the discussion section.
- c) The results (spectra) of the GC and MS measurements could be presented in a supplementary material and thus secure the identification.
Response: GCMS chromatogram has now been included as a supplementary material
Comments on the presentation and description
- d) The results of the cytotoxic measurements can be included somewhat more intensively in the results and in the discussion.
Response: The results of the cytotoxic measurement has been elaborated and also included in the discussion
- e) The further investigations are clearly and also clearly presented in the understanding of the reviewer. However, the "red thread" is somehow missing in the presentation. The authors are therefore encouraged to connect the individual steps more pointedly in the description. Thus, they allow the readers a somewhat better understanding of their train of thought when conducting the study.
Response: Additional statements have now been included in the results, discussion and conclusion section of the revised manuscript.
Minor comments:
f.) Line 24: Ocimum basilicum in italics
Response: Thank you for pointing out. In the revised version, we have corrected it.
g.) Please use capital “L” for liter throughout the manuscript.
Response: Thank you for pointing out. In the revised version, we have corrected it.
h.) Line 157: “stigmasterol” without capital “S”
Reply: Thank you for pointing out. In the revised version, we have corrected it.
j.) Line 168: Two fullstops?
Response: Thank you for pointing out. In the revised version, we have corrected it.
j.) Line 239: “~01” or “~01.”?
Response: Thank you for pointing out. Now included approximately instead of “~01.
k.) Line 270: Please use “°” for the Temperature und not superscript “0”.
Reply: Thank you for pointing out. In the revised version, we have corrected it.
l.) Line 275 and also in some other lines: Please use “h” for hours. (Not “hours” or “hrs”.)
Response: Thank you for pointing out. In the revised version, we have corrected it.
Reviewer 2 Report
Article Anti-dengue activity of phytosterol-rich extract of Ocimum basilicum L. to investigation of lipophilic (hexane) extract of Ocimum basilicum L. steam - its chemical composition and activity against dengue and chikungunya.
I don't think the name of article is quite correct. The article does not compare the content of phytosterols in the extract with other plants and no action was taken to enrich the extract with phytosterols. This is a common lipophilic extract.
Authors use only stem as part of plant for extract obtaining. It should be explained and substantiated.
It would be interesting to compare data on the chemical composition of other extracts and essential oil from Basil, which also have antiviral activity.
In addition, the authors carry out similar experiments with lipophilic extracts of other plants, in particular, Rajesh K. Joshi, Shivankar Agarwal, Poonam Patil, Kalichamy Alagarasu, Kingshuk Panda, Cherish Prashar, Mahadeo Kakade, Kusuma S. Davuluri, Sarah Cherian, Deepti Parashar, Kailash C. Pandey, Subarna Roy, Effect of Sauropus androgynus L. Merr. on dengue virus-2: An in vitro and in silico study, Journal of Ethnopharmacology, Volume 304, 2023, https://doi.org/10.1016/j.jep.2022.116044. The design of the studies is similar. Docking studies are also carried out, so the degree of novelty of the material presented is not clear.
Also in my opinion it would be good to compare such investigation each other. Perhaps about which substances are more effective.
Author Response
Response to reviewer’s comments
We appreciate the reviewer’s encouraging, critical, and constructive comments on this manuscript. The comments have been extensive and helpful in improving the manuscript. We strongly believe that the comments and suggestions have increased the scientific value of the revised manuscript by many folds. We have taken them fully into account in the revision. We are submitting the revised manuscript with the suggestion incorporated in the manuscript. The manuscript has been revised per the reviewer’s comments and suggestions. The changes have been made in the manuscript with a different color. Our responses to all the comments are as follows:
Comments and Suggestions for Authors
Reviewer #2: Reviewer's Comments
Comment 1: I don't think the name of article is quite correct. The article does not compare the content of phytosterols in the extract with other plants and no action was taken to enrich the extract with phytosterols. This is a common lipophilic extract.
Response 1: As suggested, we have modified the title.
Comment 2: Authors use only stem as part of plant for extract obtaining. It should be explained and substantiated.
Response 2: We also tested leaves extract, which did not show significant results. Hence stem extract is reported.
Comment 3: It would be interesting to compare data on the chemical composition of other extracts and essential oil from Basil, which also have antiviral activity.
Response 3: The other extracts showed antiviral activity now included in the article (Reference no 25 to 30)
Comment 4: In addition, the authors carry out similar experiments with lipophilic extracts of other plants, in particular, Rajesh K. Joshi, Shivankar Agarwal, Poonam Patil, Kalichamy Alagarasu, Kingshuk Panda, Cherish Prashar, Mahadeo Kakade, Kusuma S. Davuluri, Sarah Cherian, Deepti Parashar, Kailash C. Pandey, Subarna Roy, Effect of Sauropus androgynus L. Merr. on dengue virus-2: An in vitro and in silico study, Journal of Ethnopharmacology, Volume 304, 2023, https://doi.org/10.1016/j.jep.2022.116044. The design of the studies is similar. Docking studies are also carried out, so the degree of novelty of the material presented is not clear.
Response 4: Both the studies have been compared in the discussion with regard to chemical composition and antiviral activity to the novelty clear.
Comment 5: Also in my opinion it would be good to compare such investigation each other. Perhaps about which substances are more effective.
Response 5: Complied with and included in the revised manuscript.
Round 2
Reviewer 2 Report
I thank the authors for their attention to my recommendations and comments.
I wonder if is there difference between docking studies results? Due to abstract of the article (Joshi RK, Agarwal S, Patil P, Alagarasu K, Panda K, Prashar C, Kakade M, Davuluri KS, Cherian S, Parashar D, Pandey KC, Roy S. Effect of Sauropus androgynus L. Merr. on dengue virus-2: An in vitro and in silico study. J Ethnopharmacol. 2022 Dec 14;304:116044. doi: 10.1016/j.jep.2022.116044. Epub ahead of print. PMID: 36528212.) you made conclusion about role of stigmasterol in antiviral action. There isnt differences if you use the same molecule as ligand and the same protein. So these investigation were done before? in this case I reccommend not to describe the same results, but cite your previous investigation
Author Response
Response to reviewer’s comment
We appreciate the reviewer’s encouraging, critical, and constructive comments on this manuscript. The changes have been made in the manuscript with a different color. Our responses to all the comments are as follows:
Comments and Suggestions for Authors
Reviewer #2: Reviewer's Comments
Comment 1: I wonder if is there difference between docking studies results? Due to abstract of the article (Joshi RK, Agarwal S, Patil P, Alagarasu K, Panda K, Prashar C, Kakade M, Davuluri KS, Cherian S, Parashar D, Pandey KC, Roy S. Effect of Sauropus androgynus L. Merr. on dengue virus-2: An in vitro and in silico study. J Ethnopharmacol. 2022 Dec 14;304:116044. doi: 10.1016/j.jep.2022.116044. Epub ahead of print. PMID: 36528212.) you made conclusion about role of stigmasterol in antiviral action. There isnt differences if you use the same molecule as ligand and the same protein. So these investigation were done before? in this case I reccommend not to describe the same results, but cite your previous investigation.
Response 1: As per suggestion the manuscript has been modified accordingly.
